Particularities of the changes in young swimmers’ body adaptation to the stimuli of physical and mental stress in sports training process

Mihailescu Liliana 1
Dubiţ Nicoleta 2
Mihailescu Liviu Emanuel 3
Potop Vladimir vladimir_potop@yahoo.com 1 3 4
1 Doctoral School of Sports Science and Physical Education, University of Pitesti , Pitesti , Romania
2 Pitesti High School with Sports Program , Pitesti , Romania
3 Departament of Physical Education and Sport, University of Pitesti , Pitesti , Romania
4 State University of Physical Education and Sport , Chisinau , Republic of Moldova
Palazón-Bru Antonio
Electronic publication date: 2021 Jun 23
Publication date: 2021
Volume: 9
Electronic Location ID: e11659
Received 2021 Jan 25; Accepted 2021 Jun 1
Copyright: ©2021 Mihailescu et al.
Copyright year: 2021
Copyright holder: Mihailescu et al.
License: This is an open access article distributed under the terms of the Creative Commons Attribution License, which permits unrestricted use, distribution, reproduction and adaptation in any medium and for any purpose provided that it is properly attributed. For attribution, the original author(s), title, publication source (PeerJ) and either DOI or URL of the article must be cited.
License URL: https://creativecommons.org/licenses/by/4.0/

Keywords: Perceived stress, Competitive anxiety, Distress, Monitoring, Planning effort volume, Biochemical control, Performance, Swimming

Funding: The authors received no funding for this work.

==============================
Background

A factor favoring the swimming performance increase is the adaptation and readaptation of body energetic and functional systems to the physical and mental stress stimuli in training and competitions. The efficiency of monitoring the young swimmers’ training is based on a precise determination of the changes in the specific adaptations. The evaluation and control of the biochemical, mental and motor changes ensure the knowledge of the particularities of body adaptation in different training stages.

Methods

Six young swimmers aged 12–16 years specialized in 100 m event participated in this study, conducted in four stages (E1–general, E2–specific, E3–pre-competitive and E4–competitive). The distress occurrence during adaptation to training and competition stimuli was studied in 3 levels: mental (Cohen & Williamson Test (CWT), Cohen Perceived Stress Test (CPST), Competition Anxiety Test (SCAT), Crăciun Test (CT)), motor (workouts monitoring, planning of means on training areas, anaerobic threshold assessment and average training speed calculation) and biochemical (blood lactate (La) and blood glucose (Glu) before and after effort—5 and 15 minutes; level of metabolic biochemical parameters, lymphocytes and blood glucose, and also hormonal parameters—norepinephrine, prolactin and cortisol—before and after competition effort).

Results

Quantity results of the mental, motor and biochemical tests were analyzed in groups; the quality results for each subject in dynamics were analyzed by comparison and correlation. Psychological tests showed increases in athletes’ mental behavior by 34% at CWT (p < 0.05), by 37.5% at CPST (p < 0.05), average stress level at SCAT and 70% stress in self-confidence at CT (p < 0.01). Biochemical tests revealed an ascending dynamics of La accumulated after specific effort, with peaks in E2 period (p < 0.05) and decreases in E3 compared to E2 (p < 0.05), revealing the adaptation to specific effort and the increase in anaerobic capacity. The Glu values decrease in pre-effort and increase in minute 5 and 15 post-effort (p < 0.05) in E2 and decrease in pre- and post- effort in E3 (p < 0.05), showing the effort impact on body and ability to recover after effort. Athletes’ individual metabolic results were 50% above maximum values, mainly post-effort (42%); hormonal results were 17% over maximum values, mainly post-effort (14%). Spearman’s correlative analysis of the induced-stress stimuli in workouts revealed 6.06% significant correlations at p < 0.05 and 9.1% strong connections in competitions: 4.67% significant correlations at p < 0.05 and 6.4% strong connections.

Conclusion

The research demonstrated that the mental and biochemical tests results correlation with the specific motor tests facilitated the correct individualization of effort orientation in training and recovery and contributed to the knowledge of the particularities of young swimmers’ body adaptation to training and competition effort.

Introduction

The adaptation and permanent readaptation of human body systems to the training stimuli is the functional spring of progress in sports performance, provided that they are optimal individually (Mellalieu et al., 2009; Mihăilescu, 2018).

The specialists of this field—Bompa (2002); Dragnea & Teodorescu-Mate (2002); Platonov (2015); Salgau (2005)—state that it is a biological necessity to reach the over compensation after performing a number of training sessions, because the processes of body adaptation are superior to the ones in the previous training. During the preparatory period, the chronic fatigue syndrome is caused by successive overloaded microcycles, with insufficient recovery periods that influence three major areas: the neuromuscular system, the metabolic system and the neuro-endocrine system. The practice of the modern sports training highlights that the optimization of the sports results achieved by the athletes adapted to intense efforts is determined by the training method based on the relationship between effort-fatigue-stress; this method is the basis of the training.

Nowadays, when the sports results in swimming events exceed the capacity of the body, it is very usual to monitor the functional state of the swimmers during a training macrocycle, to analyze the physiological, kinematical and performance changes, to identify the contributions of the technique and energetic, bioenergetic and anthropometric profiles (Bădescu & Galeru, 2009; Dias, Marques & Marinho, 2012; Tucher et al., 2019; Zacca et al., 2018; Zacca et al., 2019). The training principles: specificity, overload, progressiveness, tempo, rhythm and adaptability used in the training of senior swimmers shall be also respected in the training of cadet (12–14 years old) and junior (14–16 years old) swimmers. Also, the swimmers aged 12 to 16 years will train in the effort areas taken over from Maglischo, (1993). Depending on the sports branch and the sports event, two categories of training sessions are needed: endurance training to improve the aerobic metabolism and sprint training to improve the anaerobic metabolism and strength (Maglischo, 1993; Salgau, 2005).

The relationship between recovery and fatigue and its impact on performance has attracted the interest of sport science for many years. An adequate balance between stress (training and competition load, other life demands) and recovery is essential for athletes to achieve continuous high-level performance (Kellmann & Beckmann, 2018; Kellmann et al., 2018).

In swimming, the workouts are dominated by the aerobic ergogenesis; the water in the swimming pool imposes a series of adaptive processes (e.g., Drăgan, 2002). The specialists focused also on the following matters: the acute biochemical and physiological responses to swimming training series performed at intensities based on the 400-m freestyle speed (e.g., Franken et al., 2018); the determination of the effect of 12 weeks of training on the critical velocity and maximal lactate steady state of elite swimmers (e.g., Machado et al., 2011); the classification, identification and follow-up of young swimmers’ performance; the biomechanical determinants during two competitive seasons; the analysis of the individual variations in each swimmer (e.g., Morais et al., 2021).

From a psychological point of view, stress can be often confused with emotions that become a source of stress in the interpersonal relationships (e.g., Godoy-Izquierdo, Sola & Godoy, 2011). In general, on the one hand, the stress is the negative side of distress, which causes decreased performances, decreased efficiency and the onset of disease (Grosu et al., 2015; Ware & Matthay, 2000); on the other hand, a positive side is described—the eustress which manifests itselfasa pleasant, dynamic, stimulating and exciting experience; people feel able to cope successfully with the demands they face and they put themselves in challenging situations that they know how to control. In this case, the stress works for us to improve our achievements (e.g., Šarotar Žižek, Treven & Čančer, 2013).

The prolactin increases in acute and rapid stress; at another occurrence of stress, the prolactin decreases or is even refractory. In chronic stress, prolactin is low and cortisone is increased. Stress hormones include: catecholamines, glucocorticoids, growth hormones, prolactin, vasopressin and endogenous opioids (e.g., Drăgan, 2002). The measurement of cortisol in saliva provides a reliable tool for investigations of hypothalamus-pituitary gland-adrenal gland axis activity and may be suitable for the psychobiological studies (Kirschbaum & Hellhammer, 1989). In competitive activity, the effects of victory and defeat on the response to testosterone and cortisol reveal percentage changes in hormone levels almost identical in both genders (e.g., Jiménez, Aguilar & Alvero-Cruz, 2012).

The interdisciplinary approach in the fields of sports training theory (e.g., Platonov, 2015), swimming (Smith, Norris & Hogg, 2002; Salgau, Iorgulescu & Creteanu, 2008; Mrakic-Sposta et al., 2015), medicine (Papadopoulos et al., 2014; Whdan, 2014) and psychology has made valuable contributions to the identification of the parameters that determine the occurrence of stress and has revealed its relationship with performance capacity both in training and competition (Anshel, 2001; Matthews, Deary & Whiteman, 2005; Souglis & Travlos, 2015; Vovkanych & Penchuk, 2015). Thus, Niemana & Wentz (2019) describe the influences that physical activity has on the immunity system of the body but also their metabolic aspects; Caracsso, Villaverde & Oltrans (2007) and Stroud et al. (2009) analyzed the neuroendocrine and cardiovascular responses tostress induced in the course of swimming training sessions and competitions. Kanaley et al. (2001), Crewther et al. (2013) and Soria et al. (2015) determined and analyzed the variation of stress hormones during the elementary physical exercises but also in performance sports; they consider that physical exercises of high intensity are a powerful stimulant for cortisol secretion.

We emphasize that one of the factors that can influence the results could be the control over the coaching styles during the training as a response factor to stress in young swimmers. An autocratic coaching style modulates the release of cortisol in both genders, affecting the motivational climate and the training experience of young elite swimmers. The direct and indirect relations between the coaching styles and the athletes’ intention to continue for pleasure the participation in the swimming are also important. Likewise, the motivational climate is another determining factor to take into account and that could affect the results. The motivational climate can have a different impact on adolescents’ responses to stress, causing both physiologic and psychological stress responses or reducing the performance-related stressors (Hogue, Fry & Fry, 2017; Jiménez et al., 2019; Kim et al., 2021).

However, we have not identified studies that address the use of a complex assessment system able to measure the mental and biochemical parameters relevant in determining the individual reactivity of swimmers to the effort in training and competitions and able to facilitate the correct management of swimmers’ individual preparation depending on stress level.

The aim of the research was to identify the particularities of young swimmers’ body adaptation to the stimuli of stress produced by the physical and mental efforts in training and competitive activities.

Materials & Methods

Participants

Six athletes aged 12 to 16 years specialized in 100 m event –different styles (3 juniors, 14-16 years: 2 boys—butterfly, breaststroke and one girl –backstroke; 3 cadets, 12-14 years: one boy—backstroke and 2 girls - front crawl stroke), with a swimming experience of more than 6 years, selected from the entire groupon performance criterion (they ranked in the top 10 in the finals of the National Swimming Championships. The swimmers had the height of 162.9 ± 3.78 cm, body mass 50.8 ± 4.24 kg, arm span 167.6 ± 4.40 cm and active mass 86.08 ± 1.00%. The individual competitive tests included in the competition calendar of the Romanian Swimming and Modern Pentathlon Federation depend on the age category: cadets 12 years (girls and boys) and 13–14 years (boys), 13 years (girls) and juniors 15–16 years (boys) and 14–15 years (girls).

The experimental study was approved by University of Pitesti Ethics Committee for the Doctoral School “Science of Sport and Physical Education” in accordance with the Ethical Standards of the Helsinki Declaration (ecbr5-03-2020). The subjects gave written consent to the study in accordance with the recommendations of the Ethical principles of psychologists (e.g., Meyer, 1988) and Biomedical Research Ethics Committees (e.g., World Health Organization, 2000). Given their age, the subjects had also a written consent signed by their parents regarding their participation to the research.

Methods

The evaluation of athletes’ mental state following up the training and competition stimuli was made by means of four standardized tests taken over from the specialized literature (Cohen, Towbes & Flocco, 1988; Cohen & Williamson, 1988; Woodman & Hardy, 2003; Judge et al., 2016; Britton, Kavanagh & Polman, 2017).

Two tests were used to measure the perception of stress regarding the stimuli associated to workouts and competitions: Test 1 (TΨ1), Cohen & Williamson Test (CWT, 1988) formed of 14 items with five response scales, for measuring the degree to which situations in a person’s life are assessed as stressful, exploring the subjective experience of stress: low level <25 points; average level 26-50 points; high level >50 points. Test 2 (TΨ2), Cohen Perceived Stress Test (CPST), formed of 10 items with five response scales, for determining the stress perceived in the last month, ranking feelings and thoughts in relation to the activities carried out: low level <13 points; average level 13-20 points; high level >20 points.

The following tests were given to determine the competition anxiety: Test 3 (TΨ3), Sport Competition Anxiety Test (SCAT)—a questionnaire specific to the competitive activity, which assesses the anxiety and includes 15 items with three response scales: low level <17 points; medium level 17-24 points; high level >24 points. Test 4 (TΨ4), Crăciun Test (CT) specific to the competitive activity, evaluates the anxiety on three levels: somatic, cognitive and self-confidence; this test is formed of nine items with four response scales: low level <4 points; average level 4–8 points; high level >8 points. The test highlights the cognitive anxiety (questions (Q)—3, 6, 8), somatic anxiety (Q—2, 4, 7) and self-confidence (Q—1, 5, 9) (e.g., Crăciun, 2012).

The method of determining the stress level by means of biochemical analyses was applied during workouts, in different periods of preparation, but also in competitions. The biochemical assessment by the technique of blood collecting from the arm was performed in basal stage (before the start of the preparatory period, 8.00 a.m.) and in the competitive period (5 min after the end of the individual event, between 08:00 and 16:00 O’clock, except for the athletes A5 and A6—between 18:00 and 21:00 O’clock), before and after effort. Venous blood was collected directly in special vacutainers for each type of analysis. The amount of blood collected for each parameter was: hematological parameters (VSH—1.6 ml), blood count (2 ml); biochemical parameters (Blood glucose and Total serum cholesterol—6 ml); hormone/endocrine markers: plasma catecholamines (epinephrine, norepinephrine, dopamine—4 ml), prolactin and cortisol (6 ml). The amount of blood collected from an athlete before effort was 19.6 ml and the same amount was collected after effort (total amount of 39.2 ml/ athlete). The basal analyses and the analyses during competition determined the metabolic parameters—lymphocytes (%) and blood glucose (mg/dl)—and the endocrine parameters: norepinephrine (mmol/l), prolactin (ng/ml) and cortisol (µg/dl) before the effort made in competition and immediately after effort, in minute 5. These biochemical parameters were selected because they showed changes above normal values after the competition effort (Soultanakis & Platanou, 2008; Papadopoulos et al., 2014; Zacca et al., 2019).

To determine the anaerobic capacity in training, the capillary blood test was used for measuring the blood lactate (mmol L −1) and blood glucose (mmol L−1) before effort and after effort, when the 8x100 m specialized swimming stroke was used (motor test—MT1), start at 150 s, with a pause of 1 min approximately; the competition events were split 2x50 m (MT2), with 5 s pause; in the 5th and 15th minute after effort for the assessment of the quality of the recovery capacity after effort. The Lactate Pro portable analyzer with Lactate Pro Test Strips was used to determine the blood lactate, which was generally less than +/- 2.0 mM over the physiological range of 1.0–18.0 mM (range of mean difference: −0.06 mM to 0.52 mM), Mean Squared Error and correction for Bias indicated that both the Edge and Xpress had low ‘total’ error (∼0–2 mM) for lactate concentrations <15 mM (e.g., Bonaventura et al., 2015); the blood glucose was determined by means of the portable analyzer Accu-Chek Performa Nano glucometer with Accu-Chek Performa test strips, Detection limit (lowest value displayed): 10 mg/dL (0.6 mmol/L) for the test strip, System measurement range: 10–600 mg/dL (0.6–33.3 mmol/L), Sample size: 0.6 µL, Test time: 5 s (Salgau, 2017; Ferreira et al., 2019; Arsoniadis et al., 2020a; Arsoniadis et al., 2020b). The time in the motor tests was measured with the help of the Professional Timer Stopwatch, Digital Sports Stopwatch with Countdown Timer, 100 Lap Memory, 0.001 Second Timing (water resistant, multi-functional stopwatch for swimming, running, training). The determination of the training times in the different effort zones was made following the T 30 test. After this test, the average per 100 m was calculated; this one is the time in the R2 effort zone—threshold resistance. After establishing the average per 100 m (R2 zone), the rest of the effort zones are calculated. The calculation modality is also described by W. Ernest Maglischo (Maglischo, 1993; Zacca et al., 2019).

Experimental design

The research was carried out throughout a training macrocycle formed of 4 mesocycles (MC). The basic values of stress perception were measured using the psychological tests 1 and 2 (TΨ1 and TΨ2); biochemical analysis was used by collecting metabolic and hormonal venous blood tests at the beginning of the training (testing –T1), after two weeks of holiday. Given that the subjects correspond to the two categories of competitive age (juniors 14-16 years and cadets 12-14 years), the planning of the effort volume was made differently (Table 1):

Table 1 Graph of mental, motor and biochemical indicators monitoring in macrocycle 2, 2015–2016 plan.

Month	Jan.	Feb.	March	April	Total	
Week
Indicators	08
15	18
24	25
31	01
07	08
14	15
21	22
28	29
06	07
13	14
20	21
27	28
03	04
10	11
17		
Stages	Jr.		General	Specific	Pre-Cp.	Cp.	-	–		
Ψ	ΨT1	ΨT2	ΨT3		ΨT4	-	–		
B: MT&C	BT1	B&MT1	B&MT1	B&MT2	B&C	-	–		
Stages	Cd.		General	Specific	Pre-Cp.	Cp.		
Ψ	ΨT1	ΨT2	ΨT3		ΨT4		
B: MT&C	BT1	B&MT1	B&MT1	B&MT2	B&C		
Days of training	Jr.		6	6	6	6	6	6	6	6	7	3	–	–	–	58	
	Cd.		6	6	6	6	6	6	6	6	6	6	6	6	7	79	
Days of rest	Jr.		1	1	1	1	1	1	1	1	0	0	–	–	–	8	
	Cd.		1	1	1	1	1	1	1	0	1	1	1	1	0	11	
In water training	Jr.		8	9	9	9	9	9	9	9	9	4	–	–	–	84	
	Cd.		8	9	9	9	9	9	9	8	8	9	9	9	8	113	
Off water training	Jr.		5	5	4	4	4	4	3	2	0	0	–	–	–	31	
	Cd.		5	3	3	3	2	2	3	2	2	2	2	2	2	33	
Total training	Jr.		13	14	12	12	13	13	12	11	9	4	–	–	–	113	
	Cd.		13	12	12	12	11	11	12	10	10	11	11	11	10	146	
Hours in water	Jr.		16	18	22	22	20	20	18	18	18	8	–	–	–	180	
	Cd.		10	18	22	24	22	24	26	24	28	24	24	15	8	269	
Hours off water	Jr.		3	2.3	2	2	2	2	2	1	0	0	–	–	–	16.3	
	Cd.		5	3	3	3	2	2	3	2	2	2	2	1	1	31	
Total of hours	Jr.		19	20.3	24	24	22	22	20	19	18	8	–	–	–	196.3	
	Cd.		15	21	25	27	24	26	29	26	30	26	26	16	9	300	
Volume km	Jr.		42	45	50	55	55	53	45	30	30	12	–	–	–	417	
	Cd.		40	45	45	50	45	50	55	50	60	50	44	34	12	580	
Notes.

ψ psychological testing

B biochemical testing

B: MT&C biochemical testing in training and competition

General stage I (E1)

Specific stage II (E2)

Pre-Cp. Pre-competitive - stage III (E3)

Cp. Competitive - stage IV (E4)

T1 beginning of the baseline research

T2 testing in end of general stage

T3 testing end of specific stage

T4 testing in competitive stage

MT motor test

Jr. Junior athlete (A1)

Cd. Cadet athlete (A4)

B&MT1 testing of La and Glu in motor test 8 × 100 m

B&MT2 testing of La (lactate) and Glu (glucose) in motor test 2 × 50 m

-stage I (E1), general: 4 weeks—junior and 5 weeks—cadets. The perception of stress in training was measured at the end of the general preparatory period (T2), using TΨ1 and TΨ2 tests and biochemical analyses by collecting capillary blood tests (La and Glu) before and after effort in the 5th and 15th minute, in parallel with the motor tests of training 8x100 m (B&MT1) (e.g., Nunes, Hanna Souza Messias & Vieir, 2020).

- stage II (E2), specific: 3 weeks—junior and 5 weeks—cadets. The perception of stress was measured at the end of the period(T3); biochemical analyses were made by collecting capillary blood tests (La and Glu) before and after effort in the minutes 5 and 15 (B&MT1), in parallel with the motor tests of training 8 × 100 m (e.g., Nunes, Hanna Souza Messias & Vieir, 2020).

- stage III (E3), pre-competitive (2 weeks). Capillary blood biochemical analyses (La and Glu) were collected before and after effort in the minutes 5 and 15 (B&MT2), in parallel with the motor tests of training 2 × 50 m (e.g., Nunes, Hanna Souza Messias & Vieir, 2020).

- stage IV (E4), competitive (1 week). Measurement of stress perception (tests 1 and 2) and competitive anxiety (tests 3 and 4) (T4), biochemical analyses were made by collecting venous blood tests (La and Glu) before and after competitive effort in the minutes 5 and 15, 100 m event, the specialized style.

Based on the results of motor tests, biochemical analyses and the physical training model recommended by Maglischo (1993), weekly models and individualized training models were developed per effort zones, at the beginning and the end of the research.

Analysis data

All statistical analysis of the study results was performed using the KyPlot, version 5.0, Data Analysis & Visualization software package (IBM Kyens Lab, Inc., Japan) and SPSS (IBM Co, Armonk, NY, USA), version 23. Descriptive statistics methods were used to calculate median, standard deviation (SD), range, Cohen’s d effect size, probability of superiority (PS) (e.g., Fritz, Moris & Richler, 2012), Cronbach‘s Alpha, Wilcoxon Signed Rank Test for Paired Data and Friedman Test with Replication were used to compare the means of the parameters studied between tests (e.g., Yoshioka, 2002). Pearson’s correlation coefficient was used to examine relations between variables. Statistical significance was set at p < 0.05 (e.g., De Winter, Gosling & Potter, 2016).

Results

The quantitative results recorded by the subjects during the research are presented, in their dynamics, in the following tables and figures, systematized on the three levels of the research: mental, motor and biochemical, in training and competitions. The qualitative aspects of the results were analyzed at the level of group of subjects (highlighting the common aspects and individual differences of the reactivity of the six subjects) but also at the individual level, interpreting the significance of the dynamics of each monitored indicators.

The results of the psychological particularities of stress perception in young swimmers under the influence of stimuli from training and competition are shown in Table 2.

Table 2 Psychological particularities of stress perception of the young swimmers under the influence of stimuli from training and competition (n = 6).

Psychological indicators	Median
± SD	Range	Cronbach’s Alpha	Cohen’s d	PS (%)	Z p-value	X2
p-value	
CWT (points)	T1#	27.5 ± 1.58	13	.545	1.66	87	2.032; .057	10.12 .018*	
T2	29.5 ± 1.64	12	.602	.47	61	2.232; .033*	
T3	32.5 ± 1.35	8	.588	.66	66	0.843; .461	
T4	36.0 ± 2.11	12	.423	.63	66	1.632; .208	
CPST (points)	T1#	16.0 ± 2.66	7	.273	2.26	94	1.992; .059	9.54 .023*	
T2	21.5 ± 2.32	6	.097	1.44	84	2.032; .057	
T3	22.0 ± 1.64	4	−2.798	.57	64	1.479; .169	
T4	23.5 ± 1.75	5	−1.401	.51	64	0.812; .498	
SCAT (points)	T4#	21.5 ± 5.91	15	.878	–	–	–		
CT (points)		A	8.5 ± 2.66	8	.845	–	–	2.213; .035*	13.17;	
	T4	B	6.5 ± 1.63	4	−.113	–	–	2.207; .035*	.004**	
		C	8.5 ± 1.79	5	.750	–	–	2.201; .036*		
Notes.

stage I (E1), general stage II (E2), specific stage III (E3), pre-competitive.

T1 beginning of the baseline research

T2 testing in end of general stage

T3 testing end of specific stage

T4 testing in competitive stage

A cognitive

B somatic

C -self-confidence; Cohen’s d effect size (Small 0.2; Medium 0.5; Large 0.8; Very large 1.3)

PS probability of superiority

Z Wilcoxon Signed Rank Test for Paired Data

X2 Friedman Test With Replication

T1# comparison with T4

T4# comparison with A, B, C

* p < 0.05.

** p < 0.01.

The results analysis of the stress perceived at CWT presented the distribution of the individual values on a three-level scale related to the low level value (25 points). At the beginning of the research at basal level (T1), the athletes recorded a perception of stress 10% over the low level. The analysis of the dynamics of the individual values at each stage revealed that all athletes had progressive increases of the perceived stress in training by 8% at T2 compared with T1 (p < 0.05) and by 12% at T3 –with T2 (p > 0.05). At the end of the research, at T4 compared with T1, the athletes showed an increase by 34% in the perception of stress (p <0.05). Regarding the adaptation of athletes’ body to stress stimuli in training and competitions, there are significant differences at p < 0.05, which confirms the effectiveness of planning the effort parameters in training.

The dynamics of the perception of the individual stress in training was made with the help of CPST. The distribution of the individual values was achieved on a three-level scale. When the research started (T1), the athletes presented do perception of the stress by 23% up to the average level of stress. The dynamics of the individual values at each stage showed that all athletes have progressive increases of stress perception in training sessions by 6% in T2 compared with T1 (p > 0.05) and by 27.5% in T3 –cu T2 (p > 0.05). At research ending, the athletes presented –in T4 compared with T1 - an increase by 37.5% of the perceived stress (p > 0.05). It refers to the concerns and feelings of the last month regarding the passing of the middle school graduation national exam. As for the athletes’ body adaptation to the stimuli of stress in workouts and competitions, significant differences are observed at p <0.05, which validates the proper management of effort parameters in training.

The assessment of the competitive anxiety was performed by using SCAT and CT, in the Junior National Championships, approximately one hour after the end of the event.

The analysis of the results regarding the perceived stress in competition (SCAT) showed the distribution of the individual values on a three-level scale. The competitive anxiety values reveal an average level of anxiety. Comparing the individual results, it was observed that two athletes (A2, A4) had low values of anxiety, other two athletes (A3, A6) had a medium level and the other two (A1, A5) –high anxiety.

The CT test was used as a basis for improving and detailing the criteria of assessment of the modern diagnosis regarding the competitive anxiety. Detailing the anxiety assessment scale in three levels, at each criterion, made possible a more in-depth knowledge of the competitive anxiety of young swimmers. The results of the analysis highlighted a competitive anxiety increased by 70% at cognitive and self-confidence; the less affected by the competition was the somatic anxiety with 54%, at the low level. The comparative analysis between the indicators of SCAT and CT tests reveals significant differences at p < 0.05 and p < 0.01 within the tests. The correlation of the individual values between the test indicators shows that at the low level of anxiety SCAT corresponds more to the self-confidence sphere, at medium level to the cognitive and somatic sphere; at high level –to the cognitive sphere. These strong connections show the influence of the competitive stress stimuli on the adaptation of athletes’ body.

Figure 1 presents the volume of the means of training in the young swimmers at the beginning (E1, general stage) and the end (E3, pre-competitive stage) of the research according to the specialized styles.

Figure 1 Volume of the means of training in the young swimmers at the beginning and the end of the research according to the specialized styles.

T.E., technique exercises; WL, workouts for legs; R1 zone, basic endurance; R2 zones, threshold resistance; S1, lactate tolerance; S2 zone, lactate production.

The comparative analysis highlights the motor particularities related to the content of the individual training plans. It was noticed the decrease of the volume by 5–6% except for the cadet swimmers (12–14 years old) specialized in front crawl stroke, who had a volume of 26% of warm up at the beginning of the research and 6% at the end of the research. In the research final part, it was found out that 11–13% technique exercises (TE) for backstroke swimmers and 6% for front crawl stroke were included in the content of the training means; these exercises were not necessary for the butterfly stroke and breaststroke swimmers, given the style characteristics. The workouts for legs were increased by 1–3%; the increase for the cadet swimmers (12–14 years old) in final period was by 21%, although they did not have these workouts initially. The means of relaxation had 6% in both stages, with small changes by ±1–3% at the end.

Concerning the distribution of the specific volume per effort zones in training, it was observed that the backstroke style swimmers (A1 and A4) were specialized in the medium and long events (400 m and 800 m freestyle and 200 m backstroke). At the beginning of the research, the athlete A1 (junior, 16-year-old) worked with 65% of the training volume, namely12% on S2 zone (lactate production) and 53% on R1 zone (basic endurance) with a decrease of 10% at the end (S2–25% and R1–31%). The athlete A4 (cadet, 13-year-old) worked 62% initially, with 31% on R1 and R2 zones (threshold resistance) and a decrease of 6% at the end (R1 and R2–28%) (Figs. 1A and 1B). The junior swimmers (A2, 15-year-old—butterfly and A3, 14-year-old - breaststroke) were specialized in 100 m and 200 m events. Both athletes worked with 66% of the training volume at the beginning and the end of the research, namely with 23% on R1 and 43% on R2 at the beginning and 29% on R1 and 37% on R2 at the end (Fig. 1C). The cadets (A5, 12-year-old and A6, 12-year-old) were specialized in the 50 m, 100 m, 200 m events—freestyle and backstroke. Both athletes worked with 68% of the training volume at the beginning of the research, namely 3% on S1 (lactate tolerance), 13% on S2 and 52% on R1; at the end their decrease was by 7%, namely12% on S1, 25% on S2 and 24% on R1 (Fig. 1D).

The results achieved by athletes in the motor tests that included standardized training exercises are shown in Table 3.

Table 3 Results of blood glucose and lactate concentration in pre- and post-effort motor tests in different stages of training (n = 6).

Stage	Motor test	Statist. Ind.	Mean times (sec)	Blood lactat (mmol L−1)	Blood glucose (mmol L−1)	
				pre-effort	post-effort 5 min	post-effort 15 min	pre-effort	post-effort 5 min	post-effort 15 min	
E1	8 × 100 m	Median;
± SD	80.4;
± 4.07	1.24;
± 0.59	6.25;
± 2.85	5.35;
± 2.04	103.5;
± 9.33	118.5;
± 23.44	105;
± 16.41	
Range	11.22	0.17	8.2	5.3	26	58	42	
Cohen’s d#	2.09	0.08	12.44	0.01	0.84	0.05	0.09	
PS (%)	56	52	100	50	72	51	52	
X2	–	12.00	4.00	
p-value	–	.002**	.135	
E2	8 × 100 m	Median;
± SD	79.01;
± 3.66	1.7;
± 0.74	10.7;
± 2.29	8.25;
± 1.91	99;
± 14.56	143.5;
± 37.02	117;
± 28.97	
Range	10.2	1.8	6.4	4.7	38	101	71	
Cohen’s d	0.31	0.05	1.33	1.29	0.21	0.87	0.69	
PS (%)	58	51	82	81	56	72	68	
X2	–	12.00	6.33	
p-value	–	.002**	.042*	
E3	2 × 50 m	Median;
± SD	71.9;
5.84	1.9;
0.44	9.1;
1.44	6.7;
0.83	97;
14.2	122;
18.15	98.5;
16.36	
Range	17.3	1.3	4	2	40	53	41	
Cohen’s d	1.99	0.11	0.65	0.58	0.40	0.52	0.44	
PS (%)	91	53	66	64	61	64	62	
X2	–	12.00	7.00	
p-value	–	.002**	.03*	
Notes.

Stage I (E1), general stage II (E2), specific stage III (E3), pre-competitive.

SD standard deviation Cohen’s d effect size (Small 0.2 Medium 0.5 Large 0.8 Very large 1.3)

PS probability of superiority

Cohen’s d# comparison E1 –E2, E2 –E3 and E1 –E3

X2 Friedman Test with Replication

* p < 0.05.

** p < 0.01.

The blood lactate (La) and blood glucose (Glu) were the metabolic biochemical indicators monitored during training sessions in order to highlight the relationship between effort-fatigue-recovery. The anaerobic capacity was monitored by the value of La (lactate). At the level of the group, an ascending dynamic of the La accumulated following the specific effort was observed, with peaks in E2 period, when both volume and intensity of the effort were high. In this period, it was programmed to increase the resistance to La accumulations, in pre-effort by 0.04 mmol L−1, after 5 min post-effort by 3.8 mmol L−1 and after 15 min post-effort by 2.63 mmol L−1(p < 0.01). In E3 there are decreases compared to E2, which reflects an adaptation to the specific effort, an increase of the anaerobic capacity by 0.09 mmol L−1 in pre-effort by 1.48 mmol L −1 after 5 min post-effort and by 1.11 mmol L −1 (p < 0.01) at 15 min post-effort. The values of Glu showed the following values in E2: decreases by two mmol L−1 in pre-effort, increases by 21.2 mmol L −1 in 5 min post-effort and by 11.3mmol L−1 in 15 min post-effort (p < 0.05). There are also decreases in E3 by 5.83 mmol L−1 in pre-effort, by 19.2 mmol L −1 in 5 min post- effort, and by 12.8 mmol L −1 in 15 min post-effort, which showed the impact of the effort on the body and on the ability to recover after effort (p < 0.05). The results of the training speed calculations highlight the increase by 0.03 m s−1 (1.24 s) at the 8 × 100 m motor test between E1 and E2 and by 0.16 m s−1 (7.29 s) at 2 × 50 m test in E3. The comparison with the values with E1, post-effort 5 min blood La and pre-effort blood Glu, the others are small. The performances of the motor tests (MT) were compared to the biochemical parameters, which allowed on the one hand the recording of the biochemical reactivity to the specific effort, respectively the adaptation of the body to effort, and on the other hand the establishment of the aerobic threshold speed and the calculation of the individualized training speeds in order to optimize the performance capacity from one training period to another.

The results of the metabolic and hormonal biochemical indicatorsduring competition are listed in Table 4. The analysis was based on the premise that any disturbance in hormones functioningcauses the diminution of energy efficiency and the decrease of body resistance to stress, which has a negative impact on the elite athletes.The metabolic and hormonal parameters were investigated in basal stage (T1), depending on effort (pre and post-effort) (T4) and mental stress of the competition for analyzing objectively the efficiency of athletes in relation to their individualized training level.

Table 4 Results of the modification of the metabolic and hormonal parameters in the 100 m specialized event in pre - and post-effort (n = 6).

Biochemical parameters	Statistics Ind.	Measuring stages	
		Basal	Pre-effort	Post-effort	
Lymphocytes (%)	Median ± SD	43.65 ± 7.94	43.7 ± 8.23	52.7 ± 5.96	
Range	22.9	22.7	16.3	
Cohen’s d#	–	.16	.73	
PS (%)	–	54	69	
X2p-value	1.333 .513	
Blood glucose (mg/dl)	Median ± SD	86 ± 9.81	99.87 ± 15.33	122.77 ± 9.56	
Range	26	38.67	20	
Cohen’s d#	–	.34	3.23	
PS (%)	–	58	94	
X2p-value	9.000 .011*	
Norepinephrine (mmol/l)	Median ± SD	3.04 ± 1.21	1.04 ± 0.75	6.27 ± 5.05	
Range	3.1	2.16	12.4	
Cohen’s d#	–	1.23	4.03	
PS (%)	–	80	100	
X2p-value	10.33 .005**	
Prolactin (ng/ml)	Median ± SD	6.97 ± 1.42	8.93 ± 4.02	13.15 ± 9.20	
Range	4.23	11.19	26.04	
Cohen’s d#	–	1.84	5.87	
PS (%)	–	90	100	
X2p-value	7.000 .03*	
Cortisol (µg/dl)	Median ± SD	12.45 ± 2.4	11.8 ± 2.37	14.25 ± 3.53	
Range	6.31	5.88	9.2	
Cohen’s d#	–	.62	1.31	
PS (%)	–	66	82	
X2p-value	8.333 .015*	
Notes.

SD SD - standard deviation Cohen’s d effect size (Small 0.2 Medium 0.5 Large 0.8 Very large 1.3)

PS probability of superiority

Cohen’s d# comparison basal stage with pre-effort and post-effort

X2 Friedman Test with Replication

* p < 0.05.

** p < 0.01.

The comparative analysis of the metabolic biochemical parametersin Lymphocytes (%) highlights an increase by 4% in pre-effort and by 13% post-effort (p > 0.05). In terms of Glu (mg/dl), the parameters in basalstage andin pre-effort are within the normalreference limits;the glucose concentrationin post-effort increases by 35% related to the basal level (p < 0.05). Regarding the hormonal biochemical indicators in Norepinephrine (mmol/l), both in basal and pre-effort period, these ones have normal values with a decrease by 46% related to the basal stage and an increase by 93% over the normal reference limit in post-effort (p < 0.01). The Prolactin hormone shows an increase by 37% in pre-effort and by 117% in post-effort; all measured values are within the normal reference limits (p < 0.05); the Cortisol hormone (µg/dl) has a decrease by 12% in pre-effort and an increase by 27% in post-effort (p < 0.05).

The individual results of the athletes showed that 50% of the metabolic resultsare above the maximum values at the 2 indicators, mainly post-effort (42%), which determined an individualized carbohydrate recovery anda new approach to effort and recovery, through changes in training intensities, in line with the demands of the competition, associated with intensifications of the post-effort recovery meant to reduce the risk of exceeding the maximum values. The individual results of the athletes showed that 17% of the hormonal results are over the maximum values at the three indicators, mainly post-effort (14%). This fact determined the rethinking of the training based on the premise that the effect of the stress on the brain depend on the stress hormones (cortisol, catecholamines, prolactin), which have the ability to cross the blood–brain barrier and impair the cognitive processes: attention, memory, decision making.

The results of the correlation between the metabolic, biochemical, mental and motor indicators in training are shown in Table 5.

Table 5 Correlation coefficient for the relationship between lactate and blood glucose at each testing moment in training and the average time at MT 8 × 100 m and 2 × 50 m, volume of effort zones, training volume, CWT i CPST (n = 6).

Rho,
Variables		MT1
(sec)	MT2 (sec)	V.E.Z.
(km)	T.V.
(km)	TΨ1 (points)	TΨ2 (points)	
		Stage	E1	E2	E3	E1	E3	E1	E3	E1	E2	E1	E2	
Blood Lactate (mmol L−1)	Pre-eff	E1	−.174			.313		.224		−.089		−.058		
E2		−.086							−.783		-.147	
E3			−.058		.851		.612					
5 min post-eff	E1	−.029			.559		.500		−.383		−.314		
E2		−.143							-.928*		-.412	
E3			−.086		.795		.500					
15 min post-eff	E1	−.086			.677		.736		−.589		−.371		
E2		−.289							-.912*		−.358	
E3			.203		.806		.627					
Blood Glucose (mmol L−1)	Pre-eff	E1	.657			.294		.059		−.294		−.657		
E2		.886*							.145		−.441	
E3			.493		.045		−.343					
5 min post-eff	E1	.486			.765		.294		−.412		−.943*		
E2		.486							−.638		−.736	
E3			−.2		.441		.147					
15 min post-eff	E1	.543			.441		−.088		−.088		−.771		
E2		.6							−.638		−.736	
E3			.232		.493		.313					
Notes.

Stage I (E1), general stage II (E2), specific stage III (E3), pre-competitive.

MT motor test

V.E.Z volume of effort zones

T.V. training volume

T ψ1 Cohen-Williamson Test (CWT)

T ψ2 Cohen Perceived Stress Test (CPST)

Rho Spearman coefficient correlation

* p < 0.05, rs = 0.886 poor correlation—0.2 moderate correlation—0.5 strong correlation—0.8.

Spearman’s correlative analysis of the stimuli of the stress induced in training sessions highlighted 6.06% significant correlations at p < 0.05 as follows: at E1—TΨ2 with Glu (post-effort, 5 min); at E2—MT1 with Glu (pre-effort) and TΨ1 with La (post-effort) at the 5th and 15th minute. Regarding the association of the correlation degree between variables at each tested stage, it was observed that 66 correlations were made, in which 75.8% have strong connections (9.1%), moderate connections (30.3%) and poor connections (36.4%) while 24.2% have very weak or non-existent connections. Thus, we find out that there is an interdependent relationship between the correlated variables determined in the training sessions and the efficiency of the implemented technology. At the level of the individual values analyzed by us, the correlations of this type determined the correct management of the effort for the permanent adaptation and readaptation of the athletes in various periods of training.

Regarding the Spearman’s correlative analysis of the competition-induced stress (Table 6), a percentage of 4.67% significant correlations were highlighted at p < 0.05, in E4: TΨ3 with TΨ4(A) and T Ψ1 with Norep. (basal); between Glu in pre-effort with Glu (basal); between Norep. with TΨ4 (CT) and the performance in competition; Cortisol with Limf. (basal); between Limf. post-effort with Prolac. (basal) and Limf. post-effort with Limf (basal). As for the association of the correlation degree between the competitive stress perception variables, the result of the competition performance and the biochemical parameters in basal, pre-effort and post effort stages, one can notice that 171 correlations were made, out of which 70.2% have poor connections (43.3%), moderate (20.5%) and strong ones (6.4%) while 29.8% have very poor or non-existent connections. In terms of effort-fatigue-recovery relationship, the significant correlations highlight the interdependence between the mental and physiological stress in competition and the biochemical data of the hormonal and metabolic parameters. Therefore, these correlations must be used to determine the physiological stress induced by the competition set as a goal and to create objective premises for the individualization of the post-competitive recovery.

Table 6 Correlation coefficient for the relationship between psychological tests and biochemical parameters at each testing moment in the specialized event of 100 m with the time in competition and the biochemical parameters of the basal testing (n = 6).

Rho,
variables	Stage 4 (E4)	Basal stage (T1)	
		TΨ1	TΨ2	TΨ3	TΨ4	Times
100 m (s)	Limf. (%)	Glu (mg/dl)	Norep. (nmol/l)	Prolac. (ng/ml)	Cort. (µg/dl)	
					A	B	C							
Var-s	E4	1	2	3	4	5	6	7	8	9	10	11	12	
TΨ1			−.132	−.319	−.308	.088	−.015	.493	−.464	.058	−.928*	.174	.377	
T Ψ2				−.174	−.294	−.808	−.308	−.464	−.116	−.203	.319	−.551	.841	
TΨ3					.899*	.551	−.174	.086	.371	−.429	.257	.2	−.143	
TΨ4	A					.588	.088	.116	.667	−.580	.203	−.029	−.232	
B						.088	.116	.667	−.580	203	−.029	−.232	
C							.667	.029	.406	−.319	−.551	−.232	
Limf. (%)	Pre-effort	−.348	−.145	.029	.174	.406	−.464	−.771	.714	−.6	.543	.257	−371	
	Post-effort	−.087	−.493	.371	.174	.754	−.638	−.257	.029	−.086	.2	.943*	−.543	
Glu (mg/dl)	Pre-effort	−.058	−.174	−.257	−.551	−.116	−.029	.2	−.771	.886*	−.029	.486	−.314	
	Post-effort	.290	.058	−.086	−.464	−.174	−.174	.371	−.943*	.714	−.314	.428	.143	
Norep. (nmol/l)	Pre-effort	.209	−.348	−.086	.058	−.116	.928*	.886*	−.257	.543	−.543	−.371	−.143	
	Post-effort	−.551	−.348	−.143	−.029	.319	−.174	−.6	.486	−.086	.6	.314	−.714	
Prolac. (ng/ml)	Pre-effort	.435	−.087	.086	−.290	.231	−.580	.143	−.714	.314	−.314	.771	.086	
	Post-effort	−.348	−.116	.486	.174	.464	−.754	−.429	−.029	−.086	.486	.771	−.314	
Cort. (µg/dl)	Pre-effort	.426	.412	−.406	−.735	−.485	−.309	.116	-.928*	.551	−.319	.203	.493	
	Post-effort	−.319	.116	.314	.232	.377	−.841	−.829	.486	−.657	.6	.429	−.086	
Notes.

T1 beginning of the baseline research

E4 competitive stage

Tψ1 Cohen-Williamson Test (CWT)

Tψ2 Cohen Perceived Stress Test (CPST)

Tψ3 Sport Competition Anxiety Test (SCAT)

Tψ4 Craˇciun test (CT)

Limf. Lymphocytes

Glu Blood glucose level

Norep. Norepinephrine

Prolac. Prolactin

Cort. Cortisol

Rho Spearman coefficient correlation

* p < 0.05, rs = 0.886 poor correlation—0.2 moderate correlation—0.5 strong correlation—0.8.

Discussion

One of the factors that favors the increase of swimming performance is the adaptation and readaptation of the body to the stimuli of physical and mental stress in training and competitions. The main objectives underlying the investigation of these adaptation particularities refer to: monitoring of mental, motor and biochemical indicators and planning the volume of training effort; measuring the perception of stress and competitive anxiety in accordance with the research stages and training periods; distribution of the specific volume by effort zones in training at the beginning and end of the research, depending on the specialized style; analysis of lactate and blood glucose concentration by evaluating the motor tests in different training stages; comparative analysis of the hormonal and metabolic biochemical parameters in competitions; correlative analysis of stimuli of the stress induced in training and competitions.

A major tool in conducting the research was monitoring the mental, motor and biochemical indicators and planning the training effort volume. There are studies that support the monitoring of the physiological, kinematic and performance changes induced by the swimming training in the regional athletes of the age group; the extrinsic factors (related to environment conditions, such as development and training program) can be improved by monitoring their implementation over time, to quantify changes and contributions of energetic, technique and anthropometric profiles across the first training macrocycle (16-week) in a traditional 3-peak swimming season (Bourdon et al., 2017; Tucher et al., 2019; Zacca et al., 2018; Zacca et al., 2019; Morais et al., 2013; Morais et al., 2014; Morais et al., 2020). It was also found out an increase of swimmers’ performance in 25 m and 50 m in the best stroke technique, at critical speed, in the biochemical and anthropometric parameters along the 28 weeks of training. All of these parameters can be used to monitor the training and performance throughout a season without using sophisticated equipment (e.g., Dias, Marques & Marinho, 2012). Modeling the progress of performances can serve as a tool of detecting the talents. The effect of growth during a summer break on the biomechanical profile of talented swimmers was also studied (Dormehl, Robertson & Williams, 2016; Moreira et al., 2014). Other studies focused on the monitoring of the swimming training program on the state of muscular oxygenation in short distance swimming (e.g., Jones, Parry & Cooper, 2018). The monitoring of stress and recovery states in elite swimmers before championships could be used as indicator of training status in athletes (e.g., Nicolas et al., 2019).

The results of the mental, motor and biochemical tests were analyzed by comparison and correlation in order to objectively monitor the individual response to the stimuli of training and competition. Psychological tests showed increases and changes in the mental behavior of athletes during training by 34% at CWT and by 37.5% at CPST; an average level of stress at SCAT and 70% stress in cognitive and self-confidence at CT were found out in competition. An important feature in the evaluation of stress was the analysis of the mental state following the stimuli from training and competition. Several works analyze the following issues: fluctuations of anxiety and self-confidence in competition, measuring these variables before, during and after competition (e.g., Butt, Weinberg & Horn, 2003); the development and validation of a sport-specific measure of cognitive and somatic trait anxiety (e.g., Smith, Smoll & Schutz, 2007); usefulness of heart rate variability (HRV) analyses as non-invasive means of quantifying the cardiac autonomic regulation during the precompetitive anxiety situations in swimmers (Blasquez, Font & Ortis, 2009); measurement of cognitive and somatic competitive anxiety and self-confidence (e.g., Crăciun, 2012); determining the relationship of competitive anxiety (cognitive and somatic) between some variables (age, weight, height and body mass index) of male elite swimmers (e.g., Nikseresht, Yabande & Rahmanian, 2017); investigating the nature of the relationship between precompetitive anxiety state (CSAI-2C), subjective (race position) and objective (satisfaction) performance outcomes, and self-rated causal attributions (CDS-IIC) for performance in competitive child swimmers (e.g., Polman et al., 2007), but also the avoidance of stress occurrence (Ware & Matthay, 2000; Grosu et al., 2015; Iurea & Safta, 2018).

Regarding the distribution of the specific volume per effort zones in training, in the case of the backstroke specialized swimmers, it was observed that the swimmer A1 (junior) worked at the beginning of the research with 65% of the training volume and a decrease by 10% at the end. The swimmer A4 (cadet) worked initially with 62% of the training volume and a decrease by 6% at the end. The junior swimmers (A2 - butterfly and A3 - breaststroke) worked with 66% of the training volume at the beginning and the end of the research. The cadet swimmers (A5 and A6) –front crawl stroke and backstroke –worked with 68% of the training volume at the beginning of the research but they had a decrease by 7% at the end. According to the individual results of the biochemical indicators, the distribution of the 6 effort zones in the weekly training plan was made as follows: R2 and R3 training zones were planned so that the swimmers have 34-56 h for recovery before using again the same zones. A series of S1 (lactate tolerance) and 2 series of S2 (lactate production) were placed at least 24 h of recovery after the Resistance series R2 and R3. The S3 (Power) was positioned before the training day in the R2 effort zone so that enough glycogen could be found in the muscles to support the fast-swimming speed (e.g., Maglischo, 1993). The results of the biochemical tests in training highlighted an ascending dynamic of the La accumulated after a specific effort, with peaks in E2 period and decreases in E3 compared with E2, which reflects an adaptation to the specific effort and an increase in anaerobic capacity. The highlighted Glu values decrease in pre-effort and increase in the 5th and 15th minute post-effort in E2; concerning E3, the values decreased in pre-effort and post-effort - the 5th and 15th minute, which showed the impact of the effort on the body and the capacity of recovery after the effort. Physical effort with an intensity over 80% can increase blood glucose level by discharging epinephrine (adrenaline), which causes the elimination of glucose from the liver. The results of the biochemical tests during workouts in parallel with the motor tests performances show that swimming training causes increases of blood glucose due to the intense efforts and stress to which the body is subjected (Kirschbaum & Hellhammer, 1989; Dimitriou, Sharp & Doherty, 2002; Afshari et al., 2014; Jiménez, Aguilar & Alvero-Cruz, 2012).

The individual metabolic results of the competing swimmers were by 50% above the maximum values, mainly after effort (42%); the hormonal results exceeded by 17% the maximum values, mostly after effort (14%). As for the results of the hormonal parameters determined before the competitive effort, they have lower values than the ones recorded in basal conditions for noreprinephrine but after the effort made in competition they increased, which showed that stress and intense effort cause increases in norepinephrine, without exceeding the normal values. Prolactin increased compared to baseline conditions; it also increased before or after competition, without going beyond normal maximum values. As for cortisol, before the competitive effort the subjects had lower values than the ones recorded in baseline conditions; after the effort made in competition, there were increases of cortisol level without exceeding the values recorded in basal conditions, which highlights the fact that the stress and intense effort led to increases in this indicator (Kanaley et al., 2001; Soria et al., 2015; Caracsso, Villaverde & Oltrans, 2007; Crewther et al., 2013).

The correlative analysis of stress-induced stimuli during training sessions revealed 6.06% significant correlations at p < 0.05 while in competitions there are 4.67% significant correlations at p < 0.05 (e.g., De Winter, Gosling & Potter, 2016). The research results determined that there are differences in the cognitive anxiety level shown by different categories of swimmers. These differences were related to the qualification level. This study also showed that there is a negative correlation between cognitive anxiety and performance (e.g., Parnabas, Parnabas & Parnabas, 2015).

An important indicator in the physical effort orientation in training was the distribution of the specific volume on effort zones in different periods of training, depending on the results of the mental, motor and biochemical tests andon the observation protocols performed. Thus, corrections were made to the training plan for each athlete, at motor, biochemical and mental level. For example, in the case of the athlete A4 (butterfly stroke) with low lactate level (5.3 mmoli/l in the preparatory period) and good results, a well-developed aerobic system was reached. Because high values of lactate of 10.3 mmol/l were recorded after the motor tests performed during the specific training period and a value of 10.9 mmol/l in the pre-competitive period, the swimming was performed in effort zones of low intensities with short pauses (aerobic effort) but using mainly the backstroke. On the basis of the notes included in the observation protocol, two weekly training sessions on Ergosim Conditions Simulator were introduced to correct mistakes because technical mistakes were identified in the movement of arms in backstroke (Salgau, 2008; Mihăilescu & Dubiţ, 2015). At mental level, psychological counseling was introduced once a week given that the swimmers manifested average values of stress caused both by family problems and demands of the specific and competitive training. In order to be able to restore energy reserves, the glucose values were measured after effort because the swimmers may also have decreases in the reserves of glycogen (which produces energy); so, if necessary, a hyperglycemic diet was recommended.

While carrying out the research, it was taken into account the observance of the four fatigue stages as presented in the vision of the physiologist Ulmeanu FC, (1967): oscillating fatigue (incomplete recovery), harmonious fatigue (specific to athletes), discordant fatigue (physical and mental fatigue), pathological fatigue (exhaustion) (Ulmeanu, 1967; Sargent et al., 2014). In the experimentalmethodological approach regarding the optimization of sports performance, it was taken into consideration the observance of the four phases of overcompensation as presented in the specialized literature (e.g., Bompa, 2002).

The results of the mental tests were in a functional relationship with the biochemical ones, which showed large accumulations of lactate in the specific period and decreases in the pre-competitive period. The dynamics of the lactate levels along different periods reflected the functional stress, corresponding to the mental one. Regardless of the type/nature of the test used (motor, mental or biochemical one) similar validity criteria must be ensured, so that the test provides useful and associative informationon the current or future performances (Smith, Norris & Hogg, 2002; Rodríguez-Zamora et al., 2012; Whdan, 2014; Mihăilescu & Dubiţ, 2015).

Performing an effective endurance training requires an exact monitoring of the changes in the aerobic and anaerobic capacity, as well as a careful evaluation of the training speeds by means of some biochemical blood tests. The blood tests can be used to improve the training in four ways: establishing the training speeds, recording the training progress, diagnosing the weaknesses of the training programs and comparing the potential of a swimmer with that potential of another swimmer in order to achieve high performances. The training volume in swimming is usually very large compared to the relatively short competition time. The training at high intensity intervals has been shown to improve the performance in a rather brief period (Sperlich et al., 2010; Arsoniadis et al., 2017; Salgau, 2017; Moser et al., 2020).

The anaerobic threshold tests measure the concentration of lactate in timed repetitions series, swimming at progressive speeds. The lactate concentration depending on the swimming speed shows the results of one of the most used blood tests (Bitang & Dulceanu, 2014; Nikitakis et al., 2019; Arsoniadis et al., 2020a; Arsoniadis et al., 2020b).

The methodology used at this age and in this sport, at the level of the individual values, was validated on the basis of the quantitative and qualitative analysis of the research results, opening opportunity windows for larger studies conducted nationally and internationally on swimming and other sport branches too.

Conclusions

The research results enable us to state that higher performance in swimming can be reached if the training is based on monitoring (of the motor, mental and biochemical indicators), planning (of the training effort volume per effort zones), assessment (of swimmers’ individual reactivity to effort) and control.

The correlation of the information obtained by biochemical investigations with the psychological profile of the swimmers reflects their biological adaptation to training and competition effort and helps to improve their competitive performance.

The research demonstrated that performance in swimming can be increased by adaptation of the young swimmers’ body to the stimuli of physical and mental stress in training and competitions.

Limitations of the study

This is pilot study; the research is experimental type but was operationalized by analyzing the individual results, fact that represents a limit of the research. The results obtained have relevance and scientific significance only at the level of the research subjects and cannot be generalized. Results must be considered as new knowledge to be confirmed or refuted in longitudinal research works. However, they have a great practical applicability for both swimming and other sports, where ensuring the effort-fatigue-recovery relationships decisive in achieving sports performance. The research analyzed the individual particularities of training and the adaptive changes to the training and competition effort of the young swimmers in different swimming stroke. The small number of subjects participating in the researches is explained by the impossibility of individualized monitoring and recording of the data that were methodologically operated by other coaches during training. That is why we appealed to only one coach of the club, who worked directly with the respective swimmers. The hormonal tests were collected according to the order of participation in competition: the first four swimmers from 08:00 to 16:00 O’clock and the last two swimmers from 18:00 to 21:00 O’clock, when the circadian cycles decreased; this must be taken into account in order to reproduce this study in the future. Because there is no data base with values of the two categories of biochemical parameters that we used, the analysis of the research results was made only against the basic benchmarks of laboratories and related to the individual dynamics of those values, throughout a training macrocycle, without reference to particularities of the values in elite athletes generally and swimmers particularly.

Supplemental Information

Supplemental Information 1 Appendix for analysis of research data

Volume effort training monitored in macrocycle 2; Psychological test results; Volume of training means per effort zones at the beginning and the end of the research; Results of the biochemical tests during training sessions in parallel with the performances of the motor tests; Dynamics of the metabolic and hormonal biochemical parameters in competitions; Correlation of the variables of research indicators in training; Correlation of the induced stress variables in competition.

Click here for additional data file.

Supplemental Information 2 Results of Cronbach’s Alpha calculations for psychological tests

Click here for additional data file.

This article is part of the planned thesis “Relationship between effort—fatigue stress, limiting or favoring factor in increasing the performance capacity” in the Doctoral School of Physical Education and Sports Science, CSUD University of Piteşti (contract no 941/2013). We express our gratitude to the following personalities who gave us a kind support for the achievement of this research: Professor Catalina Poiana, PhD, Senior medical specialist of endocrinology, Head of Endocrinology Department—‘C.I. Parhon” National Institute of Endocrinology of Bucharest; Prorector of “Carol Davila” University of Medicine and Pharmacy of Bucharest; Professor Graziela Vajiala, PhD—“Spiru Haret” University of Bucharest, Biochemistry discipline; Scientific researcher 1st rank; President of ANAD România; President of Council of Europe Advisory Group on Educational Issues; Daniela Mihai, teacher, PhD student, LPS Pitesti, Chemistry Discipline, Deputy Director. We also thank the lecturer Mihai Ilie, PhD and the athletes who participated in this research for their agreement, support and collaboration.

Additional Information and Declarations

Competing Interests

Author Contributions

Human Ethics

Data Availability

The authors declare there are no competing interests.

Liliana Mihailescu and Nicoleta Dubiţ conceived and designed the experiments, performed the experiments, analyzed the data, prepared figures and/or tables, authored or reviewed drafts of the paper, and approved the final draft.

Liviu Emanuel Mihailescu performed the experiments, prepared figures and/or tables, and approved the final draft.

Vladimir Potop performed the experiments, analyzed the data, prepared figures and/or tables, authored or reviewed drafts of the paper, and approved the final draft.

The following information was supplied relating to ethical approvals (i.e., approving body and any reference numbers):

The experimental study was approved by University of Pitesti Ethics Committee for the Doctoral School “Science of Sport and Physical Education” in accordance with the Ethical Standards of the Helsinki Declaration (ecbr5-03-2020).

The following information was supplied regarding data availability:

The raw measurements are available in the Supplemental Files.

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
