# Peer review of "Particularities of the changes in young swimmers’ body adaptation to the stimuli of physical and mental stress in sports training process"

_PeerJ, doi:10.7717/peerj.11659_

## Round 0.1 · original submission · Major Revisions

Several major issues have been highlighted by the reviewers and you should assess them in a revised version of the text. Please, see the comments below so as to have more information.

·

Basic reporting

The general conception of the study is particularly good, the topic is important and current. I highlight the great difficulty of conducting longitudinal studies in swimming, with multidimensional analysis, throughout a training season. On the other hand, there are several issues that need to be corrected, better explained, or presented to facilitate the understanding of the methods, the results and the discussion. Specially the paper organization should be redone.

Experimental design

The study is well structured, with an appropriate design. The small sample size is limiting, as the
authors assume, but studies with this design, in swimming, are hardly performed with much larger sample size.

Validity of the findings

The format of the tables and figures, as the authors carried out, makes the evaluation difficult.
The results present a lot of text, there is little objectivity, and the tables and figures are not selfexplanatory, so evaluating the validity of the findings is difficult. However, it seems that they agree with the literature of the area.

Additional comments

Abstract: there is no background, only the objectives.
Please do not use the term "styles" when referring to the different strokes, but prefer "strokes", for example: backstroke, breastroke, front crawl stroke, fly stroke. This is repeated throughout the article In the blood, the concentration of lactic acid is not measured, but of lactate. Biochemically, lactic acid, produced within muscle fibers, is rapidly reduced to lactate, and is thus concentrated in the blood.
CWT, CPST, SCAT, CT are abbreviations that were not previously defined in the abstract. This prevents them from being understood. Adaptation of lactic anaerobic capacity is related to the increase of lactate concentration for the
same or greater swimming speed. Reducing the lactate concentration to the same, or greater speed is aerobic adaptation. This is not clear in the abstract.
Please do not mix methods with results: when describing statistical procedures, do not reference them in the results.
There is no space between number and % the correct is 5%, not 5 %.

Introduction:
The following references, studies with longitudinal analyzes in swimming, are necessary to better introduce the study and, after discuss the results:
1. Dias, P., Marques, M., Marinho, D. (2012). Performance evaluation in young swimmers
during 28 weeks of training. Journal of Physical Education and Sport, 12, 30-38.
2. Franken, M., Mazzola, P.N., Dutra-Filho, C.S. et al. Acute biochemical and
physiological responses to swimming training series performed at intensities
based on the 400-m front crawl speed. Sport Sci Health 14, 633–638 (2018).
https://doi.org/10.1007/s11332-018-0472-z
3. Machado, M., Júnior, O., Marques, A., Colantonio, E., Cyrino, E., De Mello, M. (2011).
Effect of 12 weeks of training on critical velocity and maximal lactate steady state in
swimmers. European Journal of Sport Science, 11, 165-170.
4. Morais, J., Garrido, N., Marques, M., Silva, A., Marinho, D., Barbosa, T. (2013). The
influence of anthropometric, kinematic and energetic variables and gender on
swimming performance in youth athletes. Journal of Human Kinetics, 39, 203-211 DOI:
10.2478 / hukin-2013-0083
5. Morais, J. E., Marques, M. C., Marinho, D. A., Silva, A. J., Barbosa, T. M. (2014).
Longitudinal modeling in sports: Young swimmers ’performance and biomechanics
profile. Human Movement Science, 37, 111–122. doi: 10.1016 / j.humov.2014.07.005
6. Morais, J., Forte, P., Silva, A., Barbosa, T., Marinho, D. (2020). Data modeling for interand intra-individual stability of young swimmers ’performance: a longitudinal cluster
analysis. Research Quarterly For Exercise And Sport, published online ahead of print, 1-
13. DOI: 10.1080 / 02701367.2019.1708235
7. Moreira, M., Morais, J., Marinho, D., Silva, A., Barbosa, T., Costa, M. (2014). Growth
influences biomechanical profile of talented swimmers during the summer break. Sports
Biomechanics, 13, 62-74.
8. Tucher, G., Castro, F. A. S., Garrido, N., Fernandes, R. (2019). Monitoring changes over a
training macrocycle in regional age-group swimmers. Journal of human kinetics, 69, 213-
223.
9. Zacca, R., Azevedo, R., Chainok, P., Vilas-Boas, J.P., Castro, F.A., Pyne, D., Fernandes, R.
(2018). Monitoring age-group swimmers over a training macrocycle: energetics,
technique, and anthropometrics. Journal of Strength and Conditioning Research, 34,
818-827.
10. Zacca, R., Azevedo, R., Ramos, V. R. Jr., Abraldes, J. A., Vilas-Boas, J. P., Castro, F. A. S.,
Pyne, D. B., Fernandes, R. J. (2019a). Biophysical follow-up of age-group swimmers
during a traditional three-peak preparation program. Journal of Strength and
Conditioning Research, Volume Publish Ahead of Print - Issue - doi: 10.1519 /
JSC.0000000000002964

From line 79 to line 83, specific and current references are required.
Lines 103 - 105: no more than three references are required for each quote. Only mention the most important ones, without repetition of content. Throughout the text, try to maintain the standard.
Try to write paragraphs of 8 to 12 lines, with 3 to 5 sentences. Exceptionally long paragraphs make it difficult to understand the topic. Try to leave only one topic per paragraph.

Materials and methods
Lines 122 - 124: In place of "juniors" and "cadets" use age in years, as this categorization varies among countries.
126: What do you mean "Sportsmen"? Are the swimmers?
139 - 144: Is this paragraph necessary? Does it add anything? Explain the methods?
168 - 170: it is not necessary to describe where the analysis was carried out, but it is necessary to explain the analysis procedure.
Lines 203 and 210: which tests are 8 x 100 m and 2 x 50 m? What are the references? Between the number and the unit there is a space: it is 100 m, not 100m (but it is 5% and not 5 %).
221: While inferential statistics were used for nonparametric data, why was the descriptive with mean and standard error of the mean? I suggest median, standard deviations, range... more suitable procedures for nonparametric data.
Results There is little objectivity in the paragraphs preceding each Table and Figure. These paragraphs should briefly present what will be presented, pointing out the main results.
234 and throughout: after describing the statistical procedures, they should not be, again, in the results.
294: The competitive event is called 400 m freestyle, not front crawl, for example.
295 and beyond: training zones have not been previously defined in the introduction or methods.

Discussion
In comparison to the results presented, the discussion is short and superficial. I strongly suggest reading the recommended references to deepen the discussion.

Conclusion
It is too long, must be just related to the objectives.

·

Basic reporting

Thank you for allowing me to review this manuscript, I have enjoyed reading it. The manuscript presents a study over different periods (i.e. baseline, training, competition ...) of a very small sample of training swimmers of both genders. the most relevant observations suggest differences in some of the control moments. The experimental design conforms to the research question proposed by the authors. However, there are some variables that could have been taken into account and It could had been consider a sufficient sample of participants for the results to have greater inferential validity.

Kirschbaum, C., & Hellhammer, D. H. (1989). Salivary cortisol in psychobiological research: an overview. Neuropsychobiology, 22(3), 150-169.

Jiménez, M., Aguilar, R., & Alvero-Cruz, J. R. (2012). Effects of victory and defeat on testosterone and cortisol response to competition: evidence for same response patterns in men and women. Psychoneuroendocrinology, 37(9), 1577-1581.

Introduction is adequate, but I would like to consider the line 96 term "the stress hormone" referring to prolactin. Prolactics is undoubtedly part of the stress response, but considering prolactics as "the stress hormone" might be incorrect. If any hormone has been traditionally indicated with this definition, it would be cortisol; also in sport (see Kirschbaum & Hellhammer, 1989, Jiménez et al., 2012).

I would also like to highlight the lack of control on the coaching styles during training as a driver of the stress response in young swimmers (Jiménez et al., 2019; Kim et al., 2021). Likewise, the motivational climate is another determining factor to take into account and that could affect the results (see Hogue et al, 2017; Jiménez et al., 2019). I would recommend including these references in the introduction and its considering as study limitation.

Hogue, C. M., Fry, M. D., & Fry, A. C. (2017). The differential impact of motivational climate on adolescents’ psychological and physiological stress responses. Psychology of Sport and Exercise, 30, 118-127.

Jiménez, M., Fernández-Navas, M., Alvero-Cruz, J. R., García-Romero, J., García-Coll, V., Rivilla, I., & Clemente-Suárez, V. J. (2019). Differences in psychoneuroendocrine stress responses of high-level swimmers depending on autocratic and democratic coaching style. International journal of environmental research and public health, 16(24), 5089.

Kim, S., Park, S., Love, A., & Pang, T. C. (2021). Coaching style, sport enjoyment, and intent to continue participation among artistic swimmers. International Journal of Sports Science & Coaching, 1747954120984054.

Experimental design

The experimental design presents an important limitation: the small sample. 3 boys and 3 girls are insufficient to infer the results. In addition to the consequent gender differences in response to stress, which have not been adequately evaluated.

Line 145 and following: Could the authors include Cronbach's alpha values ​​for each of the psychometric tools?

Line 159: When the authors refer to the "baseline period", could they elaborate more clearly what they consider to be the "baseline period"? A day without physical activity? What time of day were the samples taken, etc.?

Line 160 and following: Could the authors clearly specify the hours of blood sampling and the differences with which they were taken in each athlete, if any? Could you also indicate if the samples "pre" before or after the warm-up in each of the situations? Especially since circadian cycles play an important role in these types of studies.

Why was a lactate sample not taken within 60 seconds of exertion? Could authors indicate devices (i.e. blood lactate & glucose) measurement error and limit detections?

Validity of the findings

I am not sure if it was the best statistical approaching. But, in any case, effect size and power effect must be included. Power effect is completely necessary with a short samples. Please, consider include effect sizes and power effects to all statistical analysis.

Table 5 and table 6 must incorporate p values for all significative correlations. (*) are not presented on tables.

Additional comments

The study is very interesting and provides information that I believe will be of interest to the readers of this journal. But such a small sample is always a problem to infer results, and caution should be the basis for discussion and presentation of conclusions and results. I ask the authors to consider my comments and include a well-developed study limitation section. I also suggest that the authors take into account the incorporation of the mentioned quotes or any others that they consider about the importance of coaching styles and motivational climate on young swimmers stress response to training and competition. Manuscript could improve substantially. Thank you for allowing me to enjoy reading your research, it is always an honor to contribute to your review.

·

Basic reporting

I am only including some basic comments here, as I decided to include the comments directly in the pdf document and upload it due to the large number.
- From my point of view, important aspects are missing in the introduction, but are addressed in the article in the discussion / conclusion section. In particular, the theoretical background on the interrelationships between recovery and fatigue is missing. Also missing are current results / references to athlete or training monitoring, which can be found, for example, in the consensus paper by Bordoun et al. (2017) or Kellmann et al. (2018).
[Bourdon, P. C., Cardinale, M., Murray, A., Gastin, P., Kellmann, M., Varley, M. C., et al. (2017). Monitoring athlete training loads. Consensus statement. Int. J. Sports Physiol. Perform. 12(Suppl. 2), S2161–S2170. doi: 10.1123/IJSPP.2017-0208
Kellmann, M., Bertollo, M., Bosquet, L., Brink, M. S., Coutts, A., Duffield, R., et al. (2018). Recovery and performance in sport: consensus statement. Int. J. Sports Physiol. Perform. 13, 240–245. doi: 10.1123/ijspp.2017-0759]
- Some of the statements in the Discussion and Conclusion belong more to the Introduction or Methods chapters.
- Data should be consistently anonymized throughout (including in the raw data)
- Abbreviations and designations are partly contradictory or not consistent. For example, four training stages are listed in this study order: E1- basic, E2-specific, E3-pre competitive, E 4- competitive. However, in the associated Table 1, 6 (Prepatory, Specific, pre-Cp., Cp. Transition, Specific) stages are listed for Juniors and 5 (Prepatory, Specific, pre-Cp., Cp., pre-Cp) stages are listed for Cadets.

Experimental design

In my view, the major deficits lie in the description of the study and the methods.
- It is not explained why two different test procedures are used to measure the same thing (measure the perception of stress: CWT and CPST, competition anxiety: SCAT and CT. from my point of view, several questions arise here:
1. what are the differences? How are the tests structured, which items are asked for?
2. how are the different scores taken into account in the evaluation or the subsequent calculations?
3. what are the (statistical) quality criteria of the tests? Are validation data available from existing studies?
4. what exactly is measured with these test procedures? In particular here the theoretical derivation is missing from my point of view.
5. what is the meaning of the 3 levels or how is the interpretation of the levels, i.e. what effects can be concluded from them on training or performance?
- How and according to which criteria were the effort zones divided? How was it checked during training that the athletes really swam the specified distance in the respective zone? When and according to which criteria or parameters was it decided whether and how the training had to be adapted? if individual calculations were made for each athlete, why are group values shown in the results? ...
- Were the anthropometric data measured only at the beginning (as described) or at each stage (E1-E4) (if so, where are the results)?

Validity of the findings

- Unfortunately, I can't really understand (even from the raw data) how the calculation of the corelations was done. Were the group means correlated in each case or were the individual values calculated and then aggregated?
- It is also not clear to me on which theoretical basis the assumption is based that mental training indicators (e.g. perceived stress) should have an influence on stature. Likewise, why there should be a correlation between the volume in training zones and the stature?
- If conclusions are drawn here about the relationship between recovery and fatigue, then this should also be derived in a more differentiated manner in the theoretical justification of the study. In particular, it should be questioned here which definition of recovery and fatigue is assumed. Depending on this, how is recovery or fatigue recorded or measured here in this study? There are numerous articles on this topic, e.g. :
o Bourdon, P. C., Cardinale, M., Murray, A., Gastin, P., Kellmann, M., Varley, M. C., et al. (2017). Monitoring athlete training loads. Consensus statement. Int. J. Sports Physiol. Perform. 12(Suppl. 2), S2161–S2170. doi: 10.1123/IJSPP.2017-0208
o Kellmann, M., and Beckmann, J. (eds). (2018). Sport, Recovery, and Performance:Interdisciplinary Insights. New York, NY: Routledge.
o Kellmann, M., Bertollo, M., Bosquet, L., Brink, M. S., Coutts, A., Duffield, R., et al. (2018). Recovery and performance in sport: consensus statement. Int. J. Sports Physiol. Perform. 13, 240–245. doi: 10.1123/ijspp.2017-0759
- The aim of the study is to analyze the individual (training) adaptations of the athletes resulting from the different training phases, therefore, from my point of view, it is not comprehensible that the data and calculations are used or presented almost exclusively group based. Therefore the discussion should focus more on individual differences and similarities than on group values or trends

Additional comments

Despite the criticism, first of all, thank you for the elaborate study and quite interesting data and results. From my point of view, however, it is necessary to define the main target more clearly and to prepare the theoretical background accordingly. If the focus is on individual adaptations, which is more than reasonable in the topic complex of recovery and fatigue, the results and analyses should also be individually based. Furthermore, the study design should be described in a consistent manner and all methodological decisions should be described in advance in a theory-based manner (e.g., according to which criteria will be swum in which effort zone). In particular, if interventions are undertaken based on the study results, the criteria according to which decisions are made should be clearly defined.

---

## Round 0.2 · Major Revisions

Some major points should be addressed in a new revised version of the text, following the reports of the reviewers. Please, take into account that you must reply in a letter how you assessed every comment of the reviewers, indicating:

1) Possible changes

OR

2) Why the suggested change does not make sense and that is why you have not edited the text.


I have analyzed the uploaded file by reviewer #3 and you have not replied to their comments. These should be addressed in that new version, too.

·

Basic reporting

It is clear the great effort that the authors made in answering all the questions indicated by the reviewers. However, one must consider the authors 'difficulty in transforming the reviewers' indications into material from their own text. For example, there was an indication of several articles, the intention is that these should be incorporated into the introduction or discussion, not just report what has been done. At the same time, it appears that the authors often only copied the reviewers' text, when it was just to exemplify what the reviewers sought to inform or to correct. The article under analysis is complex, at the same time interesting, but the authors must make an effort to leave the text, at the same time, objective, informative and current.

Experimental design

The design is correct, but it is too long. Please describe just the essentials, in this way, the study can be replicate elsewhere.

Validity of the findings

As mentioned before, the sample size is a great limitation considering the validity of the findings. I really think the authors should consider this as a "pilot study".

Additional comments

Please be more consistent in writing, eliminating information that is not really necessary to understand the work.
Very long sentences in English do not work. Try to rewrite in two or three sentences (example: lines 57 to 60).
Along all the text, please revise language and structure.
Line 64: "Otherwise, the stress makes adaptation impossible, the overtraining occurs leading..." It is impossible or hard to make positive adaptations?
Line 89: Is immersion bradycardia necessary for this article?
Lines 111 - 114: It is exactly as one reviewer has written.
Line 128: why is it in the singular first person?
Line 131: Is it Method?
144: Frrestyle is the competitive event, the swimmers are specialists in front crawl stroke.
146: It is body mass (kg), not weight (N).
174: "The individual competitive tests included in the competition calendar " please rewitte, it is not clear.
161: It should be in the Statistical method.
219-222: Why is there an objective here?
Results:
This section, as the study has many variables, is too long. I recommend to reduce the session with less text. Why are there references in this session?
There are six tables in the results, many are difficult to interpret. Consider the possibility of reviewing their format and check if all of them really should be presented or if there is no possibility of presenting more figures as a result. In addition, try to present each Table in a more objective way, without leaving the text repetitive or heavy.
Discussion: I do suggest a short paragraph with the main objetives and summarizing the main results found instead the first and second paragraph (lines 443 - 460).
Avoid repeat information regarding methods: 461 - 469.
470 - 487: This discussion is intersting, but there is no reference regarding this issue.
Line 496: please review the format of the reference.
I miss a lot of comparisons with other studies, as those indicated in the first review, to clarify the results and to improve the discussion quality.
Conclusion
Please be more objetive.
l

·

Basic reporting

The manuscript presents a meticulous study on the adaptation processes in young swimmers that is very interesting to improve the current academic knowledge regarding the theory and practice of high competition training. I would like to point out that the manuscript has improved substantially and that all the recommendations that were proposed by the reviewers have been correctly addressed and taken into account by the authors. I have reviewed the new final document, and consider that the current quality of the manuscript is sufficient to consider its publication in PeerJ.

I would like to thank the authors for the elegance and good work developed to meet the recommendations of the reviewers. A scientific publication must be the product of the compilation of the work of the researchers and the reviewers of the final document. Science is above all of us and our obligation is to strive to publish quality work. This manuscript comes from this common effort by authors-reviewers, which confers a good document that will awaken the interest of sports sciences. Thanks to the authors for your understanding and consideration of the recommendations.

The manuscript is ready to be published, therefore, I leave the final decision to the editor.

Experimental design

It has been substantially improved.

Validity of the findings

It has been substantially improved.

Additional comments

Dear Authors:

First of all, I want to thank you for your effort and grace in answering the questions raised by the reviewers. The reviewers' job is to collaborate with the authors to improve the final document. Thanks to your excellent consideration this work has been very easy for the reviewers. Second, I want to personally congratulate you on your research work. Sports science must be increasingly based on applicable scientific concepts. And with jobs like yours, we can be closer to this final goal. Kind regards, and let me say that I have very much enjoyed reading your manuscript and the changes you have made to it.

A cordial greeting.

---

## Round 0.3 · Minor Revisions

There are some minor changes suggested by one of the reviewers, which you should address in a new revised version of the text.

·

Basic reporting

The authors made a great effort to improve the aspects indicated in the previous evaluations. All questions raised have been resolved or answered. The quality of the article has improved significantly.

Experimental design

Properly worded.

Validity of the findings

Limited by sample size, but can be interpreted as initials.

Additional comments

Congratulations on the conduct of the research and on the way of receiving the questions brought by the reviewers.

·

Basic reporting

The efforts to implement the different requirements of the reviewers are clearly visible and well done.

Experimental design

The methods and the study design are transparent and comprehensible as far as possible

Validity of the findings

As already in the first review, from my point of view, an individually based presentation of the data and results would have been desirable. However, I also understand the authors' dilemma that many additional figures would then have been necessary and that different views of the reviewers have to be taken into account.
Perhaps a consideration for the future is to go into more detail on the individual aspects in another article based on selected parameters.
Alternative statistical methods that take greater account of individual differences could then be considered for this purpose (e. g. Bakdash et al. 2017; https://doi.org/10.3389/fpsyg.2017.00456).

Additional comments

- Line 134-137: Delete parenthesis mentioning coaching styles, as it is sufficient to know here that coaching has an impact on athlete motivation. Otherwise, the different coaching styles would also have to be explained.
- Line 136-137: why "artistic swimming sport"
- Line 299-300: name the athletes or abbreviations: "...results, it was observed that 2 athletes (A1, A4) had low values of anxiety, other..."
- Line 312-313 / Figure1: the figure shows the training volume on average per effort zone at the beginning and end of the study. But what do the mean values refer to (training unit, study phase, ...)? This would have to be added to the description.
- Some sentences are too long (e.g. line 418-422), try to shorten or split into two sentences
- Table 6: Please complete the description or legend and match it with the label in the table. For example, only TΨ1 is explained, TΨ2-4 is no longer explained. What do the variables (1-12) in the first column stand for? Overall, the table is very complex and difficult to understand.

---

## Round 0.4 · accepted · Accept

All the reviewers' concerns have been correctly addressed.

·

Basic reporting

see previous review

Experimental design

see previous review

Validity of the findings

see previous review

Additional comments

All questions and comments were clarified or implemented by the authors. Well done.